# Covering Arrays ML HPO for Static Malware Detection

**Fahad T. ALGorain** \*,† [ID] **and John A. Clark** \*,† [ID]

Department of Computer Science, University of Sheffield, Sheffield S10 2TN, UK
* Correspondence: ftalgorain1@sheffield.ac.uk (F.T.A.); john.clark@sheffield.ac.uk (J.A.C.)
† These authors contributed equally to this work.

**Abstract:** Malware classification is a well-known problem in computer security. Hyper-parameter optimisation (HPO) using covering arrays (CAs) is a novel approach that can enhance machine learning classifier accuracy. The tuning of machine learning (ML) classifiers to increase classification accuracy is needed nowadays, especially with newly evolving malware. Four machine learning techniques were tuned using cAgen, a tool for generating covering arrays. The results show that cAgen is an efficient approach to achieve the optimal parameter choices for ML techniques. Moreover, the covering array shows a significant promise, especially cAgen with regard to the ML hyper-parameter optimisation community, malware detectors community and overall security testing. This research will aid in adding better classifiers for static PE malware detection.

**Keywords:** cAgen; combinatorial testing; covering arrays; machine learning; static PE malware detection; hyper-parameter optimisation; grid search





## 1. Introduction

### 1.1. Malware and Its Detection

Malicious software is any programme that can be executed that is intended to cause harm. Academic and commercial research and development into malware detection has been a constant focus for some time now [1] and malware remains one of the most important concerns in contemporary cybersecurity. There are three approaches to malware detection: static, dynamic, and hybrid detection. Static malware detection analyses malicious binary files without executing them; this is the focus of this paper. Dynamic malware detection uses the features of run-time execution behaviour to identify malware. Hybrid detection combines the two previous approaches. Many companies and universities have significantly invested in developing new methods for identifying malware and many researchers have looked into the possibility of using machine learning (ML) to detect it.

### 1.2. Ml-Based Static Malware Detection Related Literature

Windows Portable Execution (PE) malware is one of the most common forms of encountered malware. Several works have explored the use of machine learning for PE malware detection, e.g., [2–4]. In [5] the authors provided a dataset (usually referred to as the Ember dataset) accompanied by various Python routines to facilitate access. They also provided baseline applications of various ML techniques to their dataset. In [6], the authors considered imbalanced dataset issues and model training duration. They also applied a static detection method using a gradient-boosting decision tree algorithm. Their model achieved better performance than the baseline model with less training time. (They used feature reduction based on the recommendation of the authors of [5].) Another approach used a subset of the Ember dataset for their work and compared different ML models [7]. Their goal was to identify malware families and their work was mainly concerned with scalability and efficiency. The proposed random forest model achieved a slightly better performance than the baseline model. In [8], the authors used a hybrid of two datasets, Ember (version 2017) and another dataset from the security partner of Meraz' 18 techno

cultural festival (IIT Behali). A feature selection method—Fast Correlation-based Feature Selection (FCBF)—was used to improve their model's performance. Thirteen features (with high variance) were selected. Several ML models (Decision Trees, Random Forest, Gradient boost, AdaBoost, Gaussian Naive Bayes) were introduced and trained. The Random Forest approach achieved the highest accuracy (99.9%) [9]. The study used the same dataset as this paper. It proposed an ensemble learning-based method for malware detection. A stacked ensemble of fully connected, one-dimensional convolutional neural networks (CNNs) performs the initial stage classification, while a machine learning algorithm handles the final stage classification. They evaluated 15 different machine learning classifiers in order to create a meta-learner. Several machine learning techniques were utilised for this comparison: Naive Bayes, Decision Tree, Random Forests, GB, K-Nearest Neighbours, Stochastic Gradient Descent and Neural Nets. The evaluation was conducted on the Windows Portable Executable (PE) malware dataset. An ensemble of seven neural networks with the ExtraTrees classifier as the last-stage classifier performed the best, achieving perfect accuracy. The model parameters were not stated.

Determining the full detection capabilities of the various methods is a tricky business, particularly when such methods are ML-based. Parameter selections for ML algorithms, for instance, are typically crucial to their performance and yet specific choices in the literature often lack convincing (or sometimes any) rationale. In this paper, we explore how to optimise the parameters of such algorithms, a process known as hyper-parameter optimisation (HPO). We specifically investigate the use of covering arrays as a way to combat the curse of dimensionality that results from Grid Search, which is the most common systematic approach used.

### 1.3. Grid Search and the Curse of Dimensionality

Grid Search is a powerful and widely used means of searching a parameter space to seek sets of values that give the best performance. Grid Search applies the full combinatoric evaluation of the cross-product of discretised parameter domains. A discretised domain is a set of 'representative' elements that 'span' the domain in some way. For example, the set $0, 5, ..., 95, 100$ can be considered to span the set comprising the integers $0..100$. The real interval $[0, 1]$ can be spanned for some purposes by the set with the elements $0.0$, $0.25$, $0.5$, $0.75$, and $1.0$.

The total number of combinations for Grid Search is the product of the (assumed finite) cardinalities of the individual discretised domains $D_i$.

$$totalCombinations = \prod_{i=1}^{n} card(D_i)$$

Grid Search can obviously give a thorough exploration of the parameter space, assuming that the individual domains are suitably discretised. However, in some areas of engineering, it is found that full combinatorial evaluation can be wasteful. For example, in software testing, a particular sub-combination coverage of parameter values can provide a very high fault detection capability. However, we do not know in advance the specific sub-combinations that will be the most revealing. Some effective means of exploring the combinatorial space is needed so that we do not incur the costs of a full grid search.

*Covering arrays* provide one such mechanism. Furthermore, the concept can be applied at different 'strengths', allowing flexibility in the thoroughness of the exploration of the search space at hand. Each discretised parameter domain has a set of values. A covering array is defined over the cross-product of the discretised domains $D = D_1 \times D_2 \times ... \times D_n$. The rows of the array denote specific tests. The columns of the array denote specific parameters. The $(i, j)$ element of the array is the value of parameter $j$ in test $i$. The Cartesian product of all parameter sets defines complete combinatorial coverage. The rows of a covering array provide a subset of that with a particular strength $t$. In a CA with strength $t$, then for any subset of t parameters, each possible t-tuple of values occurs in at least one row (test). This is often called t-way testing. Orthogonal arrays (OAs) are the optimal version of

CAs where each t-tuple occurs *exactly* once (rather than at least once). For some problems, an OA may not actually exist. Pairwise testing ($t = 2$) is widely used. Furthermore, it has been found more generally that small values of $t$ can actually give a strong performance in fault-finding. As $t$ increases, the size of the covering array also increases. The test set reduction achieved by covering arrays compared with a full combinatorial grid search may be very significant. The concept of strength is illustrated below.

We will generally denote a covering array with cross-product D as above and with strength $t$ by $CA_t^D$. Consider a combinatorial search space with the (discretised) parameter domains $A = \{0, 1\}$, $B = \{0, 1\}$, and $C = \{0, 1\}$, i.e., with a cross-product $D_{abc} = A \times B \times C$. A $CA_1^{D_{abc}}$ provides a suite of cases where each value of each domain occurs at least once. This is easily achieved by an array with just two rows (i.e., two cases) as shown below. $A = 0$ occurs in the first row and $A = 1$ occurs in the second. This is similar for $B$ and $C$.

| A | B | C |
|---|---|---|
| 0 | 0 | 0 |
| 1 | 1 | 1 |

If we had, say, 26 binary domains $A, B, ..., Z$, then a similar covering array, i.e., a $CA_1^{D_{abc..z}}$ with two rows would satisfy the $t = 1$ strength requirement, i.e., with rows as shown below.

| A | B | C | .. | X | Y | Z |
|---|---|---|----|---|---|---|
| 0 | 0 | 0 | .. | 0 | 0 | 0 |
| 1 | 1 | 1 | .. | 1 | 1 | 1 |

A $CA_1^D$ clearly gives a rather weak coverage (exploration) of the domain space for most purposes. In the $A, B, ..., Z$ example, only 2 from $2^{26}$ possible row values are sampled. For a $CA_2^D$, each combination of values from any two ($t = 2$) domains is present in the array. A $CA_2^{D_{abc}}$ for the A, B, and C example is given below.

| A | B | C |
|---|---|---|
| 0 | 0 | 0 |
| 0 | 1 | 1 |
| 1 | 0 | 1 |
| 1 | 1 | 0 |

We can see that the four possible values of (A, B) are present, i.e., $(A, B) = (0, 0)$ in row 0, $(0, 1)$ in row 1, $(1, 0)$ in row 2, and $(1, 1)$ in row 3. Similarly, we can see that four possible values of $(A, C)$ and the four values of $(B, C)$ are also present. Thus, all pairs of values from any two domains from A, B, and C are present and so the given array is indeed a $CA_2^{D_{abc}}$. The simplest $CA_3^{D_{abc}}$ array would give full combinatorial coverage, i.e., with all eight $(A, B, C)$ combinations, with the usual binary enumeration of 0–7 for the rows, i.e., $[0, 0, 0]$ through to $[1, 1, 1]$.

*1.4. Generating Covering Arrays*

The actual generation of arrays is not our focus. A good deal of theoretical and practical work has been carried out into algorithms to do so. Our motivation to use covering arrays was inspired by their use in software testing. The in-parameter order (IPO) method for generating $CA_2^D$ arrays for test suites is given in [10]. As they state, "For a system with two or more input parameters, the IPO strategy generates a pairwise test set for the first two parameters, extends the test set to generate a pairwise test set for the first three parameters, and continues to do so for each additional parameter."

CAs have become widely used in the combinatorial testing field where they provide a means of reducing the number of tests needed in comparison to exhaustive combinatorial testing. This has led to an increased use of a specific instance of the IPO strategy called in-parameter-order-general (IPOG). IPOG can be used to generate covering arrays of arbitrary

strengths [11]. It is a form of greedy algorithm and might not yield the test suites of minimal size. It has been noted that providing an optimal covering array is an NP-complete problem [12].

The IPOG strategy has gained traction in the software testing field. This is due to the competitive test suites that are yielded by the covering arrays it generates in comparison with other approaches for generating test suites. Additionally, it exhibits a lower generation time than other algorithms. The main goal of IPOG is to minimise the generated test suite size. This is a significant area to explore, especially when the cost of testing is very high. The duration of test cycles will be reduced with fewer tests. However, there are some cases when the test execution is very fast and does not impact the overall testing time. Instead, optimising test suites can be very costly as the test generation time can become dominating [13,14]. Optimisation of the IPOG family was introduced by [15].

There are many problems for CAs [16], where the construction of optimal values is known to be the hardest [17]. Various methods for generating covering arrays have been proposed. These include the automatic efficient test generator (AETG) system [18], deterministic density algorithm (DDA) [19,20], in-parameter order [21], and the advanced combinatorial testing system (ACTS) [22], each with its own advantages and disadvantages. Interested readers are referred to [18–21,23], respectively, for more information. The in-parameter-order (IPO) strategy grows the covering array column by column, adding rows as needed to ensure full t-way coverage. Various kinds of research on improving the covering array generation with the in-parameter-order strategy have been conducted. The original aim of the strategy was the generalisability of generating covering arrays of arbitrary strength [11] resulting in the in-parameter-order-general (IPOG) algorithm. In [10] a modification to IPOG resulted in smaller covering arrays in some instances and faster generation times. In [24], a combination of IPOG with a recursive construction method was proposed that reduces the number of combinations to be enumerated. In [25], the use of graph-colouring schemes was proposed to reduce the size of the covering arrays. In [26], IPOG was modified with additional optimisations aimed at reducing don't-care values in order to have a smaller number of rows. Most of these presented works primarily aimed to reduce the size of generated covering arrays. The FIPOG technique was shown to outperform the IPOG implementation of ACTs in all benchmarks and improved test generation times by up to a factor of 146 [15].

In this paper, we use an implementation of FIPOG provided by the cAgen tool. We show that the use of FIPOG's covering arrays can achieve excellent results for the hyper-parameter optimisation of ML-based classifiers (and better than using the default parameters) far more quickly than when using full grid search. Below, we describe the cAgen tool [27], which implements the FIPOG technique.

*1.5. The Cagen Toolset*

The cAgen toolset provides a means of generating the covering arrays and is available online [27]. This allows the user to specify parameters and sets of associated values. For technical reasons that are concerned with our specific approach to the use of covering arrays, we will assume that a discretised parameter domain with R elements is indexed by values 0, 1, (R-1). Figure 1 shows a completed specification for the $(A, B, C)$ example above.

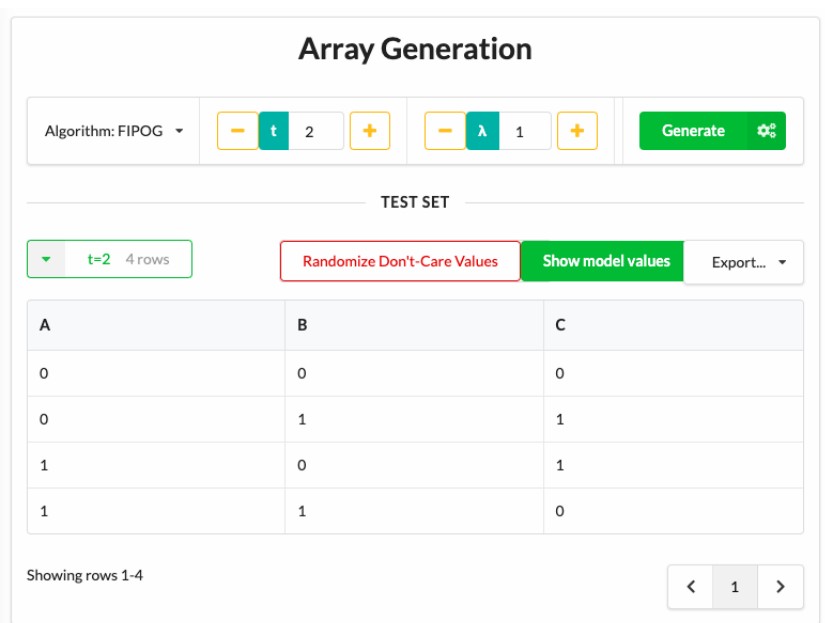

**Figure 1.** Full parameter specification for the ABC example (no constraints)

Having specified the parameters, we can invoke the generation capability of cAgen, as Figure 2 shows the array generation stage for the A, B, and C examples, where a value of t = 2 was selected. If we wanted each pair to occur multiple times, we could specify a larger value of *lambda*. Several generation algorithms are available. Figure 2 shows that we have chosen FIPOG for better performance and fast generation [27]. The array can then be stored in a variety of formats. We chose to use the CSV format throughout.

**Figure 2.** Array generation for ABC example above with t = 2.

*1.6. Array Indexing*

We use lists to represent parameter spaces. A list's elements will be either actual parameter values or else a list representing a subdomain. The values $0, 1, ..., (R - 1)$ are interpreted as indices to the corresponding elements in the discretised domain list. For example, $MAX\_DEPTH = [5, 10, 15, 20, None]$ would be a simple list with four specific integer values and a 'None' value. $LEARNING\_RATE = [0.001, 0.01, 0.1, 0.2]$ is a simple list of four real values. $MAX\_LEAVES = [[1, 2, 3], [4, 5, 6], [7, 8, 9]]$ is a list of lists of values. Here, the list cardinalities are given by $card(MAX\_DEPTH) = 5$, $card(LEARNING\_RATE) = 4$ and $card(MAX\_LEAVES) = 3$. Thus, for the list of lists, the cardinality is the cardinality of the highest-level list. Covering array values are indices to the top-level list.

Python lists allow us to include different types of elements. Thus, in $MAX\_DEPTH$, we see that integer values, as well as a 'None' parameter value, can be specified. 'None'

typically means that the algorithm can proceed as it sees fit, with no direction from the user for this parameter. ScikitLearn's ML algorithms often have such parameters as defaults. Where a parameter is represented by a simple list, then the covering array value for the parameter is used to index the specific element of the list. Thus, an array value of 2 in the covering array column corresponding to $MAX\_DEPTH$ corresponds to a parameter value of 15, i.e., $MAX\_DEPTH[2] = 15$. The indexed array element may be a list. Thus, a covering array value of 2 for $MAX\_LEAVES$, gives $MAX\_LEAVES[2] = [7, 8, 9]$. In such a case, a value is randomly selected from the indexed list $[7, 8, 9]$. Thus, each of the values 7, 8, and 9 are now selected with a probability of 0.333. In practice, we represent regular integer ranges of integer values more compactly, via the use of low, high, and increment indicators. Thus, we will typically represent the list $[1, 2, 3, 4, 5]$ by $[low, high, incr] = [1, 5, 1]$. We adopt the convention of both low and high being included in the denoted range. We distinguish between simple lists with three elements and 'compact' lists with the same three elements (selection is resolved by different routines, determined at the set-up time by the user).

### 1.7. Structure of the Paper

In this paper, we investigate whether the clear efficiency benefits a covering array approach can be brought to bear on the ML-based static malware detection problem. Section 2 describes our methodology. Section 2.2 details the performed experiments. The results are presented in Sections 3 and 4 concludes our paper.

## 2. Methodology

### 2.1. Overall Approach

We apply a variety of ML techniques and specify suitable domains for the parameters we wish to experiment with (other parameters assume defaults). We evaluate over the full combinatorial domain (for Grid Search) and over all rows of the covering arrays of interest (for t = 2, 3, 4). A full combinatorial evaluation or a full covering array evaluation (i.e., all rows evaluated) will be referred to as a 'run' or 'iteration'. We carry out 30 runs for each array of interest and for the full combinatorial case. We do this in order to gain insight into the distribution of outcomes from the technique. Some runs may give better results than others, even if the same array has been used as the basis for the run. This is due to the stochastic selection of elements within selected ranges as indicated above. Pooling the results from the 30 runs provides a means of determining an accurate and useful distribution for the approach. In practice, a user may simply use one run of a covering array search, if they are confident that it will give good enough results. Our evaluation activities aim to determine whether such confidence is justified.

### 2.2. Experimental Details

Our work uses two powerful toolkits: scikitLearn [28] and the cAgen tool [27]. The experiments are carried out using the Windows OS 11, with 11 Gen Intel Core i7-11800H, with a 2.30 GHz processor, and 16 GB RAM. The work uses a dataset [29] built using PE files from [30]. The dataset has 19,611 labelled malicious and benign samples from different repositories (such as VirusShare). The samples have 75 features. The dataset is split into a training dataset and a testing dataset (80% training, 20% testing) and can be found in [29]. All results are obtained using Jupyter Notebook version 6.1.0 and Python version 3.6.0.

A small amount of pre-processing is carried out on the malware dataset. The 'Malware' feature records the label for the supervised learning. From the remaining (i.e., input) feature columns, we restrict ourselves to binary and numerical features and so drop the 'Name', 'Machine', and 'TimeDateStamp' features. The filtered input features are then subject to scaling via scikitLearn's StandardScalar fit_transform function. The same approach is taken for all the ML approaches considered. No further feature engineering is performed. This is deliberate. Our aim is investigate ML model hyper-parameters; we wish to keep other factors constant (researchers whose focus is any of the specific ML approaches are free to engage in further optimisations should they so wish).

We evaluate a covering-array-based hyper-parametrisation on three well-established ML approaches (Decision Trees (DTs) [28], xgboost [31], and Random Forest (RF) [28,32]) together with a state-of-the-art approach (LightGBM [33]). Table 1 shows the implementation details for all four ML models using the cAgen tool. In particular, the hyper-parameter ranges of interest are shown for each technique, together with the corresponding IPM values (giving possible indices into the top level array list). The ML evaluation metric is *accuracy* as implemented by scikitLearn [34]. All three-element lists in our experiments are compact lists. The results are processed using SciPy's 'descriptive statistics' method [35].

**Table 1.** ML Models cAgen configurations.

| ML Algorithms | Hyper-Parameters | Hyper-Parameter IPM Values | T-Strengths Values | IPM Values | Number of Iterations |
|---|---|---|---|---|---|
| RF | n_estimators, max_depth, criterion, min_samples_split, min_samples_leaf, max_features | [[100, 300, 50], [350, 550, 50], [600, 800, 50]]<br>[[1, 10, 1], [11, 15, 1], [16, 20, 1],None]<br>['entropy', 'gini']<br>[[5, 25, 5], [30, 50, 5]]<br>[[5, 25, 5], [30, 50, 5]]<br>['auto', 'sqrt', 'log2', 'None'] | T-2, 3, 4 | 0, 1, 2<br>0, 1, 2, 3<br>0, 1<br>0, 1<br>0, 1<br>0, 1, 2, 3 | 30 |
| LightGBM | num_leaves, boosting_type, Subsample_for_bin, Is_unbalance, max_depth | [[20, 80, 20], [100, 160, 20]]<br>['GBDT', 'GOSS']<br>[[1000, 5000, 1000], [6000, 10000, 1000], [11000, 15000, 1000]]<br>[True, False]<br>[1, 5, 10, 15, 20, 25] | T-2, 3, 4 | 0, 1<br>0, 1<br>0, 1, 2<br>0, 1<br>0, 1, 2, 3, 4, 5 | 30 |
| Xgboost | Min_child_weight, gamma, max_leaves, reg_alpha, max_depth | [1, 2, 4, 6, 8, 10, 12, 14]<br>[[1, 4, 1], [5, 8, 1]]<br>[2, 4, 6, 8, 10, 12]<br>[0.01, 0.1, 0.2, 0.3, 0.4, 0.5]<br>[1, 5, 10, 15, 20, 25] | T-2, 3, 4 | 0, 1, 2, 3, 4, 5, 6, 7<br>0, 1<br>0, 1, 2, 3, 4, 5<br>0, 1, 2, 3, 4, 5<br>0, 1, 2, 3, 4, 5 | 30 |
| DT | max_depth, criterion, min_samples_split, min_samples_leaf, max_features | [[1, 10, 1], [11, 15, 1], [16, 20, 1], None]<br>['entropy', 'gini']<br>[[5, 25, 5], [30, 50, 5]]<br>[[5, 25, 5], [30, 50, 5]]<br>['auto', 'sqrt', 'log2', 'None'] | T-2, 3, 4 | 0, 1, 2, 3<br>0, 1<br>0, 1<br>0, 1<br>0, 1, 2, 3 | 30 |

## 3. Results

The results of hyper-parameter optimisation based on covering arrays (with strengths of 2, 3, or 4) and grid search are shown in the following tables. The best-performing parameter values are given, together with the time taken to complete the corresponding search, coverage (number of evaluations), and summary accuracy data. Tables 2–5 show results for RF, LightGBM, Xgboost, and DT, respectively. The results for Grid Search (over the same discretised parameter ranges) are also shown in each table. In the tables "No. of evaluations" is equal to the number of rows (i.e., combinations) in the covering array multiplied by the number of iterations (30).

**Table 2.** RF Model cAgen Results Comparison.

| ML Algorithms | Optimal Values | T-Values/ Grid Search | Time to Complete | No. of Evaluations | Score (min/max Accuracy and Mean) |
|---|---|---|---|---|---|
| RF | 400<br>14<br>entropy<br>5<br>10<br>None | T2 | 2 h 35 min 21 s | 480 | minmax = (0.9031205384458495, 0.9904140322251682) mean = 0.9743610662231913 |
| | 700<br>None<br>entropy<br>5<br>10<br>None | T3 | 7 h 31 min 17 s | 1500 | minmax = (0.8916989598205181, 0.9902100754640016), mean = 0.9760138690597593 |
| | 150<br>18<br>entropy<br>5<br>10<br>None | T4 | 11 h 58 min 4 s | 2880 | minmax = (0.8949622679991842, 0.9902100754640016), mean = 0.975666825299703 |
| | 650<br>19<br>entropy<br>5<br>5<br>None | Full Grid Search | 2 Days, 23 h, 58 min and 18 s | 11520 | minmax = (0.8853763002243524, **0.9906179889863349**), mean = 0.9755531264871848 |

**Table 3.** LightGBM Model cAgen Results Comparison.

| ML Algorithms | Optimal Values | T-Values/ Grid Search | Time to Complete | No. of Evaluations | Score (min/max Accuracy and Mean) |
|---|---|---|---|---|---|
| LightGBM | 60<br>gbdt<br>9000<br>False<br>25 | T2 | 1 h 41 min 18 s | 540 | minmax = (0.9726697940036713<br>0.9908219457475015),<br>mean = 0.9858956345699156 |
| | 60<br>gbdt<br>15000<br>False<br>25 | T3 | 3 h 4 min 23 s | 1080 | min,max = (0.9702223128696716<br>0.9908219457475015)<br>mean = 0.985933215491649 |
| | 40<br>gbdt<br>1000<br>False<br>15 | T4 | 5 h 36 min 55 s | 2160 | min,max = (0.9704262696308382,<br>**0.9912298592698348**),<br>mean = 0.9859165023681646 |
| | 140<br>goss<br>1000<br>False<br>20 | Full Grid Search | 7 h 57 min 14 s | 4320 | min,max = (0.9696104425861717,<br>0.9910259025086682),<br>mean = 0.985890630075313 |

**Table 4.** Xgboost Model cAgen Results Comparison (Kaggle dataset).

| ML Algorithms | Optimal Values | T-Values/ Grid Search | Time to Complete | No. of Evaluations | Score (min/max Accuracy and Mean) |
|---|---|---|---|---|---|
| Xgboost | 1<br>1<br>10<br>0.4<br>20 | T2 | 1 h 9 min 21 s | 1440 | min,max = (0.9763410157046706,<br>0.9902100754640016),<br>mean = 0.9856683549754118 |
| | 2<br>1<br>2<br>0.01<br>25 | T3 | 5 h 37 min 32 s | 8640 | min,max = (0.9763410157046706,<br>0.9902100754640016),<br>mean = 0.985969474471412 |
| | 2<br>1<br>12<br>0.1<br>25 | T4 | 23 h 12 min 33 s | 51840 | min,max = (0.9763410157046706,<br>0.9906179889863349),<br>mean = 0.9859859987460435 |
| | 1<br>1<br>10<br>0.01<br>15 | Full Grid Search | 1 d 12 h 21 min 51 s | 103680 | min,max = (0.9763410157046706,<br>**0.9906179889863349**),<br>mean = 0.9856382316161938 |

**Table 5.** DT Model cAgen Results Comparison.

| ML Algorithms | Optimal Values | T-Values/ Grid Search | Time to Complete | No. of Evaluations | Score (min/max Accuracy and Mean) |
|---|---|---|---|---|---|
| DT | Entropy<br>20<br>None<br>5<br>25 | T2 | 42.5 s | 480 | min,max = (0.744034264735876<br>0.9836834591066694)<br>mean = 0.9656014174994901 |
| | Entropy<br>11<br>None<br>5<br>15 | T3 | 1 min 28 s | 960 | minmax = (0.7352641240057108,<br>0.9840913726290027),<br>mean = 0.9662880719287512 |
| | Entropy<br>None<br>None<br>5<br>10 | T4 | 2 min 46 s | 1920 | min,max = (0.7352641240057108,<br>0.9840913426290027),<br>mean = 0.9662736249915017 |
| | Gini<br>None<br>None<br>10<br>20 | Full Grid Search | 5 min 43 s | 3840 | min,max = (0.7352641240057108<br>**0.9849071996736691**)<br>mean = 0.9660357285505473 |

We can see that the DT classifier in Table 5 is the fastest of all ML models. Even if we look at the grid search, it is still efficient with this particular technique, taking only 5 min and 43 s to finish 3840 evaluations. However, cAgen is much more efficient with only 42.5 s to finish. Although only 480 evaluations with $t = 2$ were made, this achieves the same accuracy as Grid Search but with less time and effort. The second fastest ML model after DT was LightGBM, which highlights covering the array capability even more. Table 3 shows

a huge disparity in time between the Grid Search and cAgen runs. The cAgen approach is faster than the Grid Search with only 1 h, 41 min and 18 s taken to complete the search, while the latter took 7 h, 57 min, and 14 s. Both reached excellent values for finding hyper-parameter choices while having higher accuracy. Both strength values $t = 2$ and $t = 3$ in LightGBM, even though they have almost the same results obtained, reached a score with different hyper-parameter values. cAgen is more efficient than Grid Search, using less time. The third ML model was RF, where cAgen runs reached the highest performing choices for $t = 2$ with 2 h and 35 min. In contrast, Grid Search took 2 days, 23 h and 58 min to complete the search. The difference between cAgen and Grid Search in Table 2 is significant evidence of the usefulness of covering arrays for hyper-parameter optimisation. Xgboost was the slowest of all models to achieve the best values. It took more computational time than the other techniques to achieve the best values for strengths $t = 3$ and $t = 4$, and even Grid Search. The figures below Figures 3–5 compare the accuracy results between the selected models ($t = 2$, $t = 3$ and $t = 4$) in a histogram. (These histograms are not normalised between techniques, i.e., the total counts may vary between techniques. However, the general distributions can be compared.)

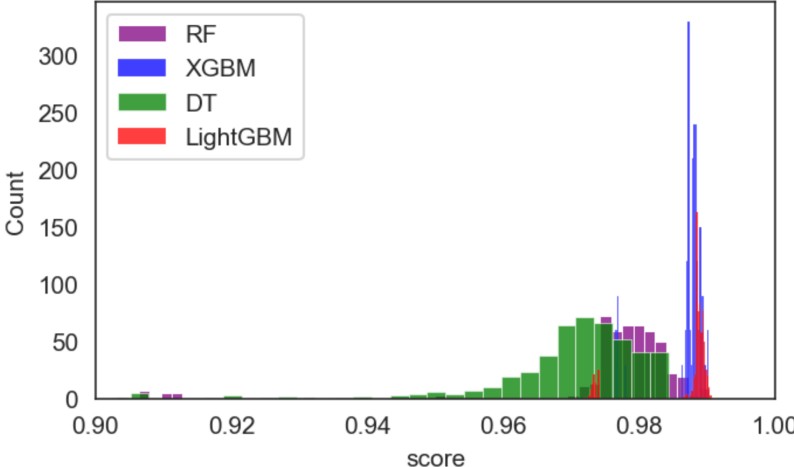

**Figure 3.** cAgen ML Models Results Comparison for Strength $t = 2$.

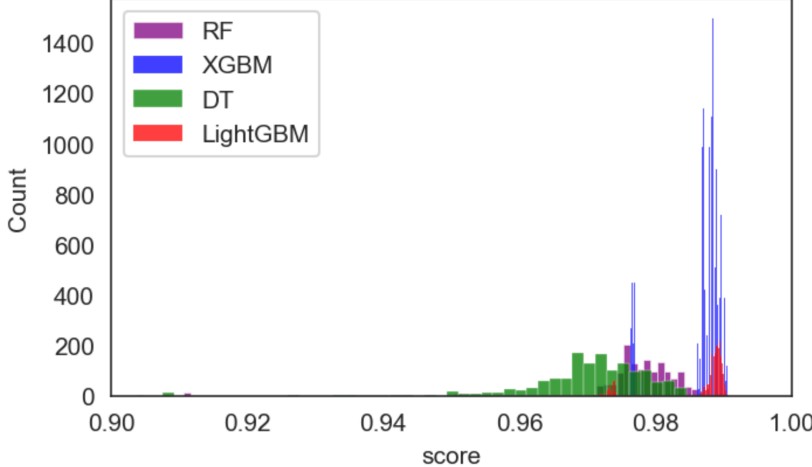

**Figure 4.** cAgen ML Models Results Comparison for Strength $t = 3$.

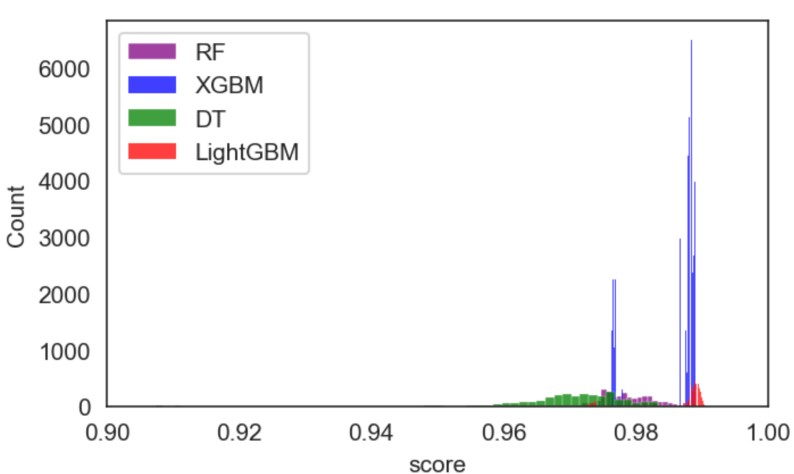

**Figure 5.** cAgen ML Models Results Comparison for Strength $t = 4$.

The authors in [9] benchmarked several ML models' performances using ensemble learning with 10-fold cross-validation. For DT and RF, the accuracy results were 0.989 and 0.984, respectively. Our model achieved 0.9849 and 0.9904. However, the main aim of our paper was to evaluate various coverage strategies, and not necessarily achieving an optimal value for each ML technique application. If explicit optima are the target, then further optimisations should be considered (see below).

### 4. Discussion

cAgen, a covering array approach with various strengths, was used to find high-performing hyper-parameters for targeted ML models. It was compared to Grid Search. Our results show that the systematic coverage offered by covering arrays can be both highly effective and efficient. The covering arrays produced by cAgen produced superior results to Grid Search across all four ML models. We highly recommend the covering arrays approach for ML researchers and the community overall. Although our work focused on improving the attained accuracy of malware classification, other security tasks may benefit from such an approach (particularly ML-based classification tasks).

For future work, we would like to assess the feasibility of adding more ML models techniques and hyper-parameters, increasing the complexity of search space/test sets, comparing different settings within the workspaces itself (e.g., FIPOG-F and FIPOG-2F), and increasing the complexity of t-way testing by adding constraints. We also believe that there is merit in considering hierarchical approaches to hyper-parametrisation, i.e., using the best values to come out of a set of runs (or even a single run) as identifying a reduced space to be systematically searched (e.g., using another covering array).

We note that we generally seek only excellent results. There is no guarantee of optimality from any of our tested approaches. An optimal result may well be given at a point that is simply not present in the cross-product of discretised domains because the discretisation process only defines *representative points* to span the domain. Furthermore, for each ML model considered, we presented what we believe are *plausible* discretised parameters ranges as the basis for our experiments. The specific choices made may affect the results. We acknowledge that other ranges are possible.

Furthermore, although one might legitimately expect higher strengths of a covering array to give rise to improved results, this is not actually guaranteed. Furthermore, after building up experience with the approaches for a specific system, one might accept that a low-strength array gives highly acceptable results very quickly, and so choose to use such arrays for all subsequent runs when training data are updated.

**Author Contributions:** Writing—original draft, F.T.A. and J.A.C.; Writing—review & editing, F.T.A. and J.A.C. All authors have read and agreed to the published version of the manuscript.

**Funding:** This research received no external funding.

**Institutional Review Board Statement:** Not applicable.

**Informed Consent Statement:** Not applicable.

**Data Availability Statement:** Dataset can be found in https://www.kaggle.com/datasets/amauricio/pe-files-malwares.

**Conflicts of Interest:** The authors declare no conflict of interest.

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
