# Peer review of "Covering Arrays ML HPO for Static Malware Detection"

_2673-4117, doi:10.3390/eng4010032_

Round 1
Reviewer 1 Report
The paper described a work on hyper-parameter optimization using covering arrays to enhance machine learning classifier accuracy. The study intends to address the problem of the curse of dimensionality in a grid search. The study used cAgen toolkit for comparing the performance of selected machine learning techniques including Random Forest, Decision Tree, LightGBM and Xgboost. Although the title of the paper contains the words "Static Malware Detection", the content does not reflect it. It is only tested on malware datasets but not specifically described how the malware was accurately detected by the method. Its results have not been explained in relation to malware detection, especially on how the methods improve detection accuracy.
The flow of the story is quite difficult to understand and lacks an explanation that demonstrates the applicability of the proposed methods with malware detection. It also contains many spelling and grammatical mistakes. The paper can be improved by explaining the proposed methods in relation to malware detection including all the parameters/features involved in the detection and keeping them consistent in the entire process of the study so that the results are useful and assist practitioners in the security area specifically the intrusion detection systems.
Author Response
Response to Reviewer 1 Comments
Point 1: The paper described a work on hyper-parameter optimization using covering arrays to enhance machine learning classifier accuracy. The study intends to address the problem of the curse of dimensionality in a grid search. The study used cAgen toolkit for comparing the performance of selected machine learning techniques including Random Forest, Decision Tree, LightGBM and Xgboost. Although the title of the paper contains the words "Static Malware Detection", the content does not reflect it. It is only tested on malware datasets but not specifically described how the malware was accurately detected by the method. Its results have not been explained in relation to malware detection, especially on how the methods improve detection accuracy.
Response 1:
The features that are used are those present in our dataset. There are 75 columns in the indicated dataset. We have added the following paragraph:
A small amount of pre-processing is carried out on the malware dataset. The ‘Malware’ feature records the label for supervised learning. From the remaining (i.e. input) feature columns we restrict ourselves to binary and numerical features and so drop the ‘Name’, ‘Machine’, and ‘TimeDateStamp’ features. The filtered input features are then subject to scaling via scikitLearn's StandardScalar fit\_transform function. The same approach is taken for all ML approaches considered. No further feature engineering is done. This is deliberate. Our aim is to investigate ML model hyper-parameters; we wish to keep other factors constant. (Researchers whose focus is any of the specific ML approaches are free to engage in further optimisations should they so wish.)
Datasets are the standard vehicle for evaluation.
Point 2: The flow of the story is quite difficult to understand and lacks an explanation that demonstrates the applicability of the proposed methods with malware detection. It also contains many spelling and grammatical mistakes. The paper can be improved by explaining the proposed methods in relation to malware detection including all the parameters/features involved in the detection and keeping them consistent in the entire process of the study so that the results are useful and assist practitioners in the security area specifically the intrusion detection systems.
Response 2:
The whole paper is concerned with the comparison of various ML techniques when applied to a specific binary classification problem (malware, benign) over a specific dataset. The dataset is a malware dataset.
We cannot explain every feature but we have indicated that we focus on numerical features only (binary is 0,1). We have significantly updated the presentation of the paper and relocated elements of the text. We believe the content and flow are now much better. Please note that the paragraph indicated above now completes the entire process of applying the ML techniques to the indicated dataset. It indicates that the same pre-processing is applied consistently. The paragraph we have added indicates that we eschew any further feature engineering. It is perfectly feasible that different optimisations could be applied in the context of the different ML models. We would wish to avoid this because of the complications it induces in evaluating our core aim, which is to understand the comparative performance of various coverage strategies.
We thank the referee for his comments which have sparked the indicated improvements in our paper and have led to a more detailed and clearer exposition of methodological rationale.

Reviewer 2 Report
The paper is compactly presented. Visible details in the figures and tables. Excellent results are purchased, but as for scientific and practical significance - optimal may be discussed more.
Author Response
Thank you for the time taken to review our paper.
Kind Regards,
Fahad ALGorain and John Clark
Reviewer 3 Report
The authors proposed a paper titled “Covering Arrays ML HPO for Static Malware Detection”, which revolves around the issue of using covering Arrays (CA) to tune the machine learning algorithms for Malware detection. The research topic is very important and the paper has an interesting findings that can be used in order to solve the problem of Malware classification.
The following minor modifications are suggested to improve the quality of the paper.
- In the “Literature Review” section, the author does not provide any overview of the previous research in malware detection. Please add the previous research in malware detection in the literature review section
- I suggest comparing the accuracy of the proposed approach with the other existing works for malware classification.
Author Response
Response to Reviewer 3 Comments
Point 1: In the “Literature Review” section, the author does not provide any overview of the previous research in malware detection. Please add the previous research in malware detection in the literature review section
Response 1:
We have extended the review work, but maintain a focus on Windows ME malware detection with an ML detection angle. This is Section 1.2.
Point 2: I suggest comparing the accuracy of the proposed approach with the other existing works for malware classification.
Response 2:
We provide a brief baseline comparison at the end of section 3. However, the real aim of the paper is to show that given a variety of ML models that need hyper-parametrising, a covering array is a good way of doing it. Comparing specific applications with the aim of achieving ‘optimality’ is not actually the goal. Also, to compare like with like it is important to compare precisely with Windows PE malware, not malware detection in general.
The referee may be interested in some more detail of our application.
A small amount of pre-processing is carried out on the malware dataset. The ‘Malware’ feature records the label for the supervised learning. From the remaining (i.e. input) feature columns we restrict ourselves to binary and numerical features and so drop the ‘Name’, ‘Machine’, and ‘TimeDateStamp’ features. The filtered input features are then subject to scaling via scikitLearn's StandardScalar fit\_transform function. The same approach is taken for all ML approaches considered. No further feature engineering is done. This is deliberate. Our aim is investigate ML model hyper-parameters; we wish to keep other factors constant. (Researchers whose focus is any of the specific ML approaches are free to engage in further optimisations should they so wish.)
